# Chiral Amphiphilic Secondary Amine-Porphyrin Hybrids for Aqueous Organocatalysis

**DOI:** 10.3390/molecules25153420

**Published:** 2020-07-28

**Authors:** Aitor Arlegui, Pol Torres, Victor Cuesta, Joaquim Crusats, Albert Moyano

**Affiliations:** 1Section of Organic Chemistry, Department of Inorganic and Organic Chemistry, University of Barcelona, Faculty of Chemistry, C. de Martí i Franquès 1-11, 08028 Barcelona, Spain; aitorarlegui@hotmail.com (A.A.); torres.yeste.pol@gmail.com (P.T.); v.cuesta@ub.edu (V.C.); 2Institute of Cosmos Science, C. de Martí i Franquès 1-11, 08028 Barcelona, Spain

**Keywords:** J-aggregates, aqueous organocatalysis, porphyrins, switchable catalysis, aldol reaction, Michael reaction

## Abstract

Two chiral proline-derived amphiphilic 5-substituted-10,15,20-tris(4-sulfonatophenyl)porphyrins were prepared, and their pH-dependent supramolecular behavior was studied. In neutral aqueous solutions, the free-base form of the hybrids is highly soluble, allowing enamine-based organocatalysis to take place, whereas under acidic conditions, the porphyrinic protonated core of the hybrid leads to the formation of self-assembled structures, so that the hybrids flocculate and their catalytic activity is fully suppressed. The low degree of chirality transfer observed for aqueous Michael and aldol reactions strongly suggests that these reactions take place under true “in water” organocatalytic conditions. The highly insoluble catalyst aggregates can easily be separated from the reaction products by centrifugation of the acidic reaction mixtures, and after neutralization and desalting, the sodium salts of the sulfonated amine-porphyrin hybrids, retaining their full catalytic activity, can be recovered in high yield.

## 1. Introduction

Catalysis is a key concept in contemporary chemistry, with far-reaching implications in several fields ranging from molecular biology to the sustainable large-scale synthesis of drugs, agrochemicals, or functional materials. While, in the past four decades, impressive advances with regard to catalytic efficiency and enantioselectivity have been achieved, we are still far from the exquisite degree of sophistication exhibited by multienzymatic catalytic systems. Actual catalytic processes in live organisms usually take place in parallel, and the required spatial and temporal control of catalytic activity often entails its modulation by a variety of external factors. It is thus now widely recognized that the introduction of stimuli-responsive behavior in artificial catalysts (“switchable catalysis”) is a key factor for the development of more efficient nature-inspired catalytic systems [1]. The modulation of catalytic activity or selectivity by the formation of self-assembled of supramolecular systems has been studied by several research groups in the past two decades [2,3,4,5,6], but the use of aggregation/dissociation processes for the reversible switching of catalytic activity has remained relatively unexplored [7,8,9,10]. Amphiphilic *meso*-(4-sulfonatophenyl)porphyrins constitute one of the simplest systems whose aggregation state can be controlled by means of an external stimulus. In aqueous solutions at acidic pH, typically at values below 4.8, the diprotonation of the central pyrroleninic core induces the formation of the so-called J-aggregates [11], stabilized by ion-pair contacts between the cationic porphyrin centers and the anionic sulfonate groups of the periphery. In the context of our research program on the use of amphiphilic porphyrins and their J-aggregates in aqueous organocatalysis [12,13], we recently found that by covalently binding a cyclic secondary amine with a 4-sulfonatophenylporphyrin scaffold we could modulate the organocatalytic properties of the hybrid structures **1** and **2** in aqueous aminocatalytic reactions that can take place through enamine intermediates (Scheme 1). In particular, we showed that the catalytic activity of the cyclic secondary amine moieties of the functionalized porphyrins **1** and **2** is indirectly regulated by varying the pH value of the medium, which controls the aggregation/deaggregation state of the porphyrin scaffold. In this way, the porphyrin can be rendered inactive at a pH value in which the cyclic amine unit would otherwise be catalytically active [14].

These amine-porphyrin hybrids can readily be prepared, albeit in relatively low overall yields, by a mixed-porphyrin synthesis [15] from pyrrole, benzaldehyde, and a suitable *N*-Boc-protected aminoaldehyde, followed by treatment of the intermediate mono-functionalized porphyrins (obtained from the statistical porphyrin mixture by chromatographic purification) with hot concentrated sulfuric acid [14]. After some experiments, we were pleased to find that the catalytic activity of the pyrrolidine moiety in the amphiphilic isoindoline–porphyrin hybrid **2** for the aldol reaction of cyclohexanone with 4-nitrobenzaldehyde could be selectively and reversibly switched on and off by adjusting the homogeneity of its solutions through pH variations [14]. To the best of our knowledge, these results constitute the first example in organocatalysis in which the activity of a catalyst can be conveniently regulated in a reversible way by modulating its aggregation state as a response to pH variations of the media.

Since the pioneering work of Barbas III [16], it is widely recognized that the enantioselectivity of aminocatalytic processes mediated by water-soluble catalysts (such as non-hydrophobic α-amino acids) is strongly diminished in aqueous solution, probably due to the disruption of the hydrogen-bond network necessary to organize the transition state, and this is the reason why the use of organocatalysts containing relatively large hydrophobic moieties is generally mandatory in order to preserve the enantiomeric excesses of the products achieved in non-aqueous solvents [17,18,19]. In fact, aqueous aminocatalytic reactions may actually be taking place by mechanisms other than homogeneous “in water” enamine activation (i.e., by general base catalysis with low molecular weight amines, or by micellar catalysis through “on water” processes) [20,21]. We thought that it would be interesting to introduce chirality in our amphiphilic secondary amine-porphyrin hybrids, in order to evaluate the degree of asymmetric induction that could be achieved in aqueous organocatalytic aldol and Michael reactions.

## 2. Results and Discussion

When we tried to obtain the 5-(*N*-Boc-2-(*S*)-pyrrolidinyl)-10,15,20-triphenylporphyrin **3** by the mixed condensation of readily available [22,23,24] *N*-Boc-(*S*)-prolinal 4 with benzaldehyde and pyrrole, we found that even after careful chromatographic purification of the crude reaction mixture, the desired compound was always obtained together with the corresponding “N-confused porphyrin” [25] **5** (Scheme 2).

We hypothesized that the competing formation of **5** was due to a destabilizing steric interaction between the pyrrole rings and the *N*-Boc-pyrrolidine moiety, and we replaced **4** by the known [26] *N*-Boc-(*S*)-homoprolinal **6** in the mixed porphyrin synthesis. To our satisfaction, no trace of “*N*-confused porphyrin” was detected, and only the “normal” porphyrin **7** was formed (along with the other expected “normal” porphyrins with a different *meso*-substitution pattern), and could be isolated in a satisfactory 10% yield (24% of the statistical theoretical yield) upon chromatographic purification. The sulfonation/deprotection of this compound proceeded without problems, and after the usual isolation/purification procedure, the disodium salt of the amphiphilic pyrrolidine-porphyrin hybrid **8** was obtained in an 82% yield (Scheme 3).

With the aim of introducing a longer spacer between the pyrrolidine and the porphyrin moieties, we synthesized the *N*-Boc-protected bicyclic aldehyde **11** by direct reductive amination [27] between 4-(hydroxymethyl)piperidine **9** [28] and *N*-Boc-(*S*)-prolinal **4**, followed by Swern oxidation of the intermediate alcohol **10** (Scheme 4).

This aldehyde was then submitted to the mixed porphyrin synthesis under Lindsey’s conditions, affording **12** in a 4% yield; finally, the sulfonation/neutralization/purification sequence gave access to the chiral amphiphilic porphyrin hybrid **13** in a 58% yield (Scheme 5).

As a previous step to ascertain the influence of the aggregation state in the organocatalytic activity of amphiphilic porphyrin hybrids **8** and **13**, we determined their inner core pK_a_ values by spectrophotometric procedures [29,30,31]. Since the first protonation of the porphyrin inner core facilitates the entrance of a second proton, the basicity of both pyrroleninic nitrogen atoms are often very similar, so that only the porphyrin (pK_a1_ + pK_a2_)/2 mean value can be measured experimentally. The pK_a_ value of the porphyrin core of **8** was found to be 3.97, and therefore this compound is ca. 4 times more acidic than **1** (pK_a_ = 4.56) [14], a fact that is probably due to the greater proximity of the positively charged nitrogen atom in the 2-methylpyrrolidinium group than in the piperidinium one. On the other hand, the pK_a_ value for **13** was 4.39, only marginally lower than that of 1, suggesting that the presence of the additional protonated pyrrolidine moiety exerts only a small destabilizing effect on the diprotonated porphyrin core beyond that of the piperidinium. It should be noted, however, that the pK_a_ obtained for **13** is only approximate, since for this compound, in the buffered solutions in which the spectra were recorded for the fully protonated species at pH = 2.60, the presence of small amounts of J-aggregate (even at very low concentrations, 10^−7^ M) precluded a completely unambiguous spectrophotometric determination (see the electronic Appendix A).

The homoassociation tendency of the fully protonated forms of the amphiphilic porphyrins can be conveniently studied by recording their UV-Vis spectra in solutions of increasing concentration under comparable conditions of aqueous HCl 0.1 M (see the electronic Appendix A). The presence of aggregation is monitored through the appearance of a red-shifted J-aggregate band, compared to that of the deprotonated monomer [32,33,34]. Comparing the critical aggregation concentrations of the acidic forms of **8** (3.78 × 10^−5^ M) and of **13** (1.75 × 10^−6^ M) to those of **1** (9.28 × 10^−7^ M) and of 2 (7.50 × 10^−6^ M) [14], in the same HCl 0.1 M media, that of 8 is between one and two orders of magnitude greater than those of the three others, which are much more similar among them. This result indicates an appreciable destabilization of the *J*-aggregate of **8**, probably due to the fact that the conformational mobility of the (2-pyrrolidinyl)methyl substituent at the 5-*meso* position hinders the packing of the monomer units. Nonetheless, the concentration of the porphyrin needed for catalysis (typically 10^−2^ M) is still about three orders of magnitude higher than the critical aggregation of the protonated hybrid **8**, assuring in this way that the catalyst can be conveniently deactivated by the acid-induced flocculation of the porphyrin.

We next turned our attention to the catalytic activity of the chiral amphiphilic porphyrins **8** and **13**, taking as a benchmark reaction the well-known [17,35,36,37,38] aldol addition of cyclohexanone **14** to 4-nitrobenzaldehyde **15** (Scheme 6 and Table 1). For comparison purposes, some of the results previously obtained with the achiral porphyrins **1** and **2 [14]** are included in Table 1.

A pH of 6.7 was measured for 0.1 M aqueous solutions of the sodium salts of amphiphilic porphyrins **1**, **2**, **8**, and **13**, showing the absence of basic hydrolysis of the amine moieties. As seen in the third entry of Table 1, at this pH, the disodium salt of **8** gave results, with respect to both yield and diastereoselectivity, similar to those of the achiral piperidine-porphyrin hybrid **1** (entry 1) for the aldol addition of cyclohexanone to 4-nitrobenzaldehyde: A 96% yield after two days at rt, 63:37 *anti* (**16a**)/*syn* (**16b**) dr. The achiral isoindoline-porphyrin hybrid **2** (entry 2) showed an almost complete *anti*-diastereoselectivity. Chiral HPLC analysis of the aldol product mixture showed that the *anti*-isomer **16a** was obtained with an almost negligible enantioselectivity (1.9% ee, favoring the “List–Houk” (2*S*,1’*R*) enantiomer). On the other hand, the *syn* diastereomer **16b** was obtained in racemic form. The catalysis of this reaction by porphyrin **13** in water was somewhat less efficient (entry 4 in Table 1), since after 5 days at rt, some starting aldehyde was still observed, and the isolated yield of the aldol adducts (70:30 *anti*/*syn* dr) was 89%. In this case, both diastereomers were obtained in optically active form, albeit with low enantioselectivity (16.7% ee for **16a**, 11.8% ee for **16b**). In both compounds, the major enantiomer also had a (2*S*) absolute configuration. As expected, when the pH of the medium was lower than 4.3, both 8 and 13 were completely inactive due to the high insolubility of the aggregates, and at pH 4.0 (entry 5 in Table 1), 13 gave no trace of product after a week of stirring at rt.

We also tested the catalytic ability of the chiral bicyclic pyrrolidine derivative **17** (the *N*-deprotected form of **10**, obtained in a 73% yield as shown in Scheme 7). In a buffer solution of pH 6.7, 17 was clearly less active than the corresponding amphiphilic hybrid **13**, since after 7 days at room temperature (rt), a 52% yield of the aldol adducts **16a** and **16b** was isolated, with a similar *anti*/*syn* dr (entry 6 in Table 1). The ees for both adducts were, however, multiplied by at least a factor of two. The addition of one equivalent of *p*-toluenesulfonic acid (entry 7 in Table 1) resulted in a substantial lowering of the catalytic activity, although the enantioselectivity was further increased (60% ee for **16a**, 32% ee for **16b**). It is thus evident that the attachment of the sulfonated porphyrin moiety, although fostering the catalytic activity of the chiral amine unit in water, brings about an important diminution of the enantioselectivity. This strongly argues in favor of a true “in water” enamine mechanism for our amphiphilic porphyrin hybrids, since the less water-soluble ligand **17** most probably acts by micellar catalysis [20].

We examined next the aqueous aldol addition of acetone and 4-nitrobenzaldehyde, a reaction that is notoriously difficult to catalyze by small water-soluble organic molecules (Scheme 8) [16,39]. While both piperidine and isoindoline were able to catalyze this reaction, the piperidine-porphyrin hybrid **1** did not show any catalytic activity, in neutral conditions; on the other hand, the use of the isoindoline-derived porphyrin **2** allowed the reaction to proceed in an excellent yield, although 7 days were needed to achieve complete conversion (entries 1 and 2 in Table 2). We were pleased to find that both chiral amphiphilic porphyrins **8** and **13** were able to catalyze this reaction, although with null (entry 3 in Table 2) or very low (entry 4 in Table 2) enantioselectivities. Amino alcohol **17** showed a low catalytic activity (entry 5 in Table 2), with no enantioselectivity.

The enamine-mediated Michael addition of cyclohexanone **14** to 2-nitrostyrene **20** was also studied in aqueous media (Scheme 9), and the best results were obtained with hydrophobic or heterogeneous catalysts [19,40,41,42,43,44,45].

We observed that addition of THF (tetrahydrofuran) to the aqueous solution in order to increase the solubility of 2-nitrostyrene in the reaction medium was necessary for the reaction. Under these conditions, only the isoindoline-derived porphyrin **2** showed a moderate catalytic activity, affording a 23% yield of the Michael adduct **21** (in a 91:9 *syn*:*anti* diastereomeric ratio) after three days at rt (entry 2 in Table 3). When we examined the catalysis of this Michael addition with the chiral porphyrin hybrids, after seven days at rt, no product was observed when **8** was used as a catalyst at neutral pH (entry 3 in Table 3). Some catalytic activity was observed with **13** (30% yield after 40 h at rt, 18% ee of the major *syn* diastereomer, entry 4 in Table 3). It should be noted that **17** again proved to be a less efficient catalyst (16% yield after 40 h), although the *syn* adduct was obtained with a good enantioselectivity (75.6% ee; entry 4 in Table 3).

Finally, it is worth noting that as previously described for their achiral analogs **1** and **2** [14], the chiral catalyst aggregates **8** and **13**, after acidification of the reaction mixtures with concentrated aqueous HCl, can readily be separated from the aldol or Michael adducts by centrifugation. The wet precipitates are carefully treated with solid sodium carbonate until a purple-colored solution, containing the trisodium salts of the porphyrin monomers, is obtained. This solution is then submitted to medium-pressure reverse-phase column chromatography. Using water as the mobile phase, inorganic salts are first eluted; increasing the gradients of methanol (from 0% to 50%) results in the subsequent elution of the porphyrins in the protonation state corresponding to neutral pH (i.e., the disodium salt of **8** and the sodium salt of **13**). Evaporation of the solvents under reduced pressure followed by lyophilization afforded the corresponding purple-colored solids, which are bench-stable for prolonged periods of time, in a ca. 80% yield.

In summary, we prepared two new chiral proline-derived amphiphilic 5-substituted-10,15,20-tris(4-sulfonatophenyl) porphyrins, and their pH-dependent supramolecular behavior was studied. In neutral aqueous solutions, the free-base anionic forms of the hybrids are highly soluble, allowing enamine-based organocatalysis to take place, whereas under acidic conditions, the porphyrinic protonated core of the hybrid leads to the formation of self-assembled structures, which are very insoluble in aqueous medium. The low degree of chirality transfer observed for aqueous Michael and aldol reactions strongly suggests that these reactions take place under true “in water” organocatalytic conditions. The precipitated J-aggregates can easily be separated from the reaction products by centrifugation of the acidified reaction crudes, and after neutralization and desalting, the anionic forms of the sulfonated amine-porphyrin hybrids, retaining their full catalytic activity, can be recovered in high yield.

## 3. Materials and Methods

### 3.1. General Methods

Commercially available reagents, catalysts, and solvents were used as received. Dichloromethane for porphyrin synthesis was distilled from CaH_2_ prior to use, and THF was dried by distillation from LiAlH_4_. Water of Millipore Q quality (18.2 MW.cm, obtained from Milli-Q1 Ultrapure Water Purification Systems, Millipore, Billerica, MA, USA) was used.

^1^H (400 MHz) NMR spectra were recorded with a Varian Mercury 400 spectrometer. Chemical shifts (δ) are given in ppm relative to the peak of tetramethylsilane (δ = 0.00 ppm); coupling constants (*J*) are given in Hz. The spectra were recorded at room temperature. Data are reported as follows: s, singlet; d, doublet; t, triplet; q, quartet; m, multiplet; br, broad signal. IR spectra were obtained in a Nicolet 6700 FTIR instrument, using ATR techniques. UV-vis spectra were recorded on a double-beam Cary 500-scan spectrophotometer (Varian); cuvettes (quartz QS Suprasil, Hellma) cm were used for measuring the absorption spectra. The porphyrin solutions in water were carefully degassed by gentle bubbling a nitrogen gas stream prior to the spectrophotometric measurement. pH measurements were performed on a CRISON Micro pH 2000 pH-meter (Crison 52-04 glass electrode) at room temperature. The pH-meter was calibrated prior to each measurement with buffers at Ph = 7.00 and 4.00 (Metrohm). Thin-layer chromatography was carried out on silica gel plates Merck 60 F_254_, and compounds were visualized by irradiation with UV light. Flash column chromatography was performed using silica gel Merck 60 (particle size: 0.040–0.063 mm).

The HPLC analyses of the sulfonated porphyrins were performed on a Shimadzu high-performance liquid chromatograph equipped with two LC-10AS pumps, a Shimadzu CBM controller, an analytical precolumn Resolve C18 (Waters), and a Nucleosil 120-5C18 analytical column (250 mm × 4 mm), using an elution gradient consisting of a mixture of methanol and tetrabutylammonium phosphate buffer (3 mmol·L^−1^; pH = 7.0) (1:1 *v:v*) to pure methanol over a period of 30 min at a flow rate of 0.6 mL min^−1^ (~2700–1050 psi). The elution profile was monitored at λ = 414 nm (UV-Vis detector SPD-6AV). Chiral HPLC analyses of aldol and Michael reaction products were performed on a Shimadzu instrument containing a LC-20-AD solvent delivery unit, a DGU-20AS degasser unit, and a SPD-M20A UV/VIS Photodiode Array detector, with chiral stationary phases (250 mm × 4.6 mm Daicel Chiralpak^®^ IC and 250 mm × 4.6 mm Phenomenex^®^ i-cellulose-5 columns; Phenomenex España, S.I., C. de Valgrande 8, Alcobendas, 28018, Madrid). All solvents were of HPLC grade and were carefully degassed prior to use. At time 0, the sample was injected.

*N*-Boc-protected aminoaldehyde **10** was obtained according to the literature procedure [26].

### 3.2. Spectrophotometric Titrations of the Porphyrins

The (pK_a1_ + pK_a2_)/2 values of the diprotonated porphyrins were determined at room temperature with an error of ±0.02 units of pK_a_ by monitoring the absorbance changes at a fixed wavelength (typically at an absorption maximum of one of the two species involved in the acid-base equilibrium) of micromolar solutions of the substance of an identical concentration of the porphyrin at different pH values, which were prepared by the addition of small volumes (~0.2 mL) of a concentrated mother solution of the free-base porphyrin in water over solutions of acetic acid—sodium acetate buffers (10 mL) of a total concentration 0.1 M. All titrations were experimentally reproducible when repeated several times, preparing in each case new fresh solutions of all the reagents. The porphyrin solutions in water were carefully degassed by gentle bubbling of a nitrogen gas stream prior to the spectrophotometric titrations. The apparent pK_a_ values were then obtained from the Henderson–Hasselbach equation by graphic interpolation using the following expression: (pKa_1_ + pKa_2_)/2 = pH + log ([diacid]/[base]), where the ratio of acid to base in each solution was calculated, at a given wavelength, as: [diacid]/[base] = (Abs_base_ − Abs)/(Abs − Abs_diacid_). All spectra used in the pK_a_ determinations showed accurate enough isosbestic points. The experimental numerical values and the regression plots for each porphyrin are presented in the electronic Appendix A.

### 3.3. Synthetic Procedures and Product Characterization

#### 3.3.1. General Procedure for the Mixed Porphyrin Synthesis

Dry dichloromethane (380 mL/mmol *N*-Boc amino aldehyde) was introduced in a round-bottomed reaction flask equipped with magnetic stirring and a reflux condenser and purged for 15 min with nitrogen. The reaction flask was charged successively with the *N*-Boc amino aldehyde (**4**, **6** or **11**, 1.0 equiv.), freshly distilled benzaldehyde (3.0 equiv.) and with freshly distilled pyrrole (4.0 equiv.). The resulting solution was stirred for 5 min at rt and boron trifluoride etherate (0.40 equiv.) was added in one portion. Stirring was maintained for 3H under nitrogen, after protecting the reaction flask from direct contact with light. At this point, *p*-chloranil (3.0 equiv.) was added, and the reaction mixture was heated to reflux for 1H in open air. After cooling to rt, most of the solvent was removed by rotary evaporation, taking care that the final volume was ca. 10 mL/mmol *N*-Boc protected amino aldehyde. This residue was submitted to chromatographic purification on silica gel, eluting with dichloromethane. A fraction containing 5,10,15,20-tetraphenylporphyrin (TPP) eluted first, followed by a more polar fraction containing a mixture of the amino-substituted porphyrins. This fraction was submitted to a second chromatographic purification on silica gel, eluting with dichloromethane/methanol mixtures of increasing polarity. In this way, the target monosubstituted porphyrin was separated from the more functionalized porphyrins. Further purification can be achieved upon recrystallization from dichloromethane/hexane [15].

*N-Boc-(S)-2-[(4-(hydroxymethyl)piperidin-1-yl)methyl]pyrrolidine* (**10**). In a 50-mL round-bottomed flask, equipped with magnetic stirring, 4-(hydroxymethyl)piperidine **9** [26] (1.96 g, 17 mmol), *N*-Boc-(*S*)-prolinal **4** [22] (1.70 g, 8.5 mmol) and anhydrous methanol (15 mL) were added sequentially, and the solution was stirred at rt for 2H under nitrogen. Next, a preformed solution of sodium cyanoborohydride (1.0 g, 8.5 mmol) and of anhydrous zinc dichloride (1.16 g, 8.5 mmol) in anhydrous methanol (15 mL) was added with the aid of a cannula, and the resulting mixture was stirred under nitrogen for 48 h at rt. A 1 M aqueous solution of NaOH (10 mL) was added in one portion, and most of the methanol was distilled off in vacuo. The remaining aqueous solution was diluted with water (20 mL) and extracted thoroughly with ethyl acetate (5 × 20 mL). The combined organic extracts were dried over sodium sulphate, and the solvent was stripped off by distillation under reduced pressure. The residue was purified by column chromatography (silica gel), eluting with 1:1 EtOAc/MeOH. In this way, the desired alcohol **10** (1.79 g, 6.0 mmol) was obtained as a yellowish oil (71% yield).

^1^H-NMR (400 MHz, CDCI_3_, TMS_int_): δ (ppm) = 4.01–3.77 (m, 1H), 3.49 (d, *J* = 6.3 Hz, 2H), 3.37–3.24 (m, 2H), 3.12–3.00 (m, 1H), 2.87–2.73 (m, 1H), 2.63–2.30 (m, 1H), 2.27–2.07 (m, 2H), 1.99–1.76 (m, 6H), 1.69 (d, *J* = 11.4 Hz, 2H), 1.46 (s, 9 H), 1.36–1.14 (m, 3H). ^13^C-NMR (100 MHz, CDCl_3_): δ (ppm) = 154.68, 68.11, 61.83, 55.34, 53.40, 46.72, 46.30, 38.71, 30.01, 29.17, 28.90, 28.70, 23.64, 22.78. HRMS (ESI^+^): *m/z* calcd. for MH^+^ [C_16_H_31_N_2_O_3_] = 299.2329, found 299.2331. [α]^20^_D_ = −39.6 (c = 0.71; CH_2_Cl_2_).

*N-Boc-(S)-2-[(4-formylpiperidin-1-yl)methyl]pyrrolidine* (**11**). To a cold (−78 °C) stirred solution of oxalyl chloride (1.0 mL, 12 mmol) in dry dichloromethane (4 mL) under an Ar atmosphere, a solution of anhydrous DMSO (0.8 mL, 12 mmol) was added via syringe. The resulting solution was stirred at −78 °C for 30 min, and after the addition of a solution of the alcohol **10** (1.79 g, 6.0 mmol) in dry dichloromethane (8 mL), stirring was maintained for 30 min at the same temperature. Triethylamine (3.3 mL, 24 mmol) was added dropwise, and the reaction mixture was stirred at −78 °C during 4H; after warming up to rt, a 10% *w/w* aqueous solution of ammonium chloride (20 mL) was added in one portion. The phases were separated, and the aqueous one was extracted with dichloromethane (3 × 10 mL). The combined organic phases were washed with aqueous-saturated sodium bicarbonate (4 × 20 mL), dried over Na_2_SO_4_, and concentrated in vacuo to afford 1.56 g (5.3 mmol, 88% yield) of the aldehyde **11** as a yellowish oil.

^1^H-NMR (400 MHz, CDCl_3_, TMS_int_): δ (ppm) = 9.62 (s, 1H), 3.97–3.73 (m, 1H), 3.37–3.20 (m, 2H), 3.05–2.85 (m, 1H), 2.81–2.62 (m, 1H), 2.58–2.33 (m, 1H), 2.30–2.00 (m, 4H), 1.95–1.75 (m, 6H), 1.73–1.54 (m, 2H), 1.44 (s, 9 H). ^13^C-NMR (100 MHz, CDCl_3_): δ (ppm) = 204.12, 154.64, 61.51, 55.31, 54.39, 52.72, 48.12, 46.29, 29.86, 28.68, 25.78, 25.61, 23.61, 22.76. HRMS (ESI^+^): *m/z* calcd. for MH^+^ [C_16_H_29_N_2_O_3_] = 297.2178, found 297.2176. [α]^20^_D_= −54.9 (c = 0.55; CH_2_Cl_2_).

*(S)-5-[(N-Boc-pyrrolidin-2-yl)methyl]-10,15,20-triphenylporphyrin* (**7**). Obtained in a 10% yield (0.17 g, 0.23 mmol) from *N*-Boc-2-(*S*)-pyrrolidinethanal **6** (0.50 g, 2.3 mmol) as a purple-colored solid. ^1^H-NMR (400 MHz, CDCl_3_, TMS_int_): δ (ppm) = 9.92–9.59 (m, 2H), 8.95 (d, *J* = 8.3 Hz, 2H), 8.81 (s, 4H), 8.26–8.15 (m, 6H), 7.83–7.70 (m, 9 H), 5.82–5.55 (m, 1H), 5.08–4.96 (m, 1H), 4.85–4.70 (m, 1H), 2.35–2.15 (m, 2H), 1.95–1.71 (m, 2H), 1.67–1.53 (m, 9 H), 1.48–1.33 (m, 2H), −2.75 (br, 2H). ^13^C-NMR (100 MHz, CDCl_3_): 143.09, 142.55, 142.49, 142.18, 142.09, 134.68, 132.93–130.54 (br, 4C), 127.78, 126.84, 126.74, 119.91, 119.84, 119.73, 118.95, 116.28, 115.74, 62.73, 47.22, 38.83, 37.98, 29.77, 28.94, 23.68, 22.83. FTIR (ATR): ν = 3318, 2920, 1655, 1594, 1465, 1396, 1163,880, 700 cm^−1^. UV-vis [CH_2_Cl_2_, λ_max_ nm (ε, L·mol^−1^·cm^−1^), 3.43 × 10^−5^ M]: 417 (400,000), 516 (14,600), 551 (7000), 591 (4500), 647 (3800). HRMS (ESI^+^): *m/z* calcd. for MH^+^ [C_48_H_44_N_5_O_2_] = 722.3495, found 722.3489.

*(S)-5-[1-N-Boc-2-(methylpyrrolidinyl)piperidin-4-yl]-10,15,20-triphenylporphyrin* (**12**). Obtained in 4% yield (95 mg, 0.28 mmol) from *N*-Boc-(*S*)-2-[(4-formylpiperidin-1-yl)methyl]pyrrolidine **11** (0.89 g, 3.0 mmol) as a purple-colored solid. ^1^H-NMR (400 MHz, CDCl_3_, TMS_int_): 9.75 (d, *J* = 4.6 Hz, 2H), 8.92 (d, *J* = 4.9 Hz, 2H), 8.81 (dd, *J* = 12.7 Hz, *J’* = 4.8 Hz, 4H), 8.21 (dd, *J* = 7.1 Hz, *J’* = 2.0 Hz, 6H), 7.86–7.67 (m, 9 H), 5.39 (m, 1H), 4.23 (m, 1H), 3.86–3.18 (m, 5H), 2.78 (m, 2H), 2.12 (m, 4H), 1.62 (d, *J* = 11.1 Hz, 9 H), 1.01–0.75 (m, 2H), −2.63 (br, 2H). ^13^C-NMR (100 MHz, CDCl_3_): 154.63, 142.52, 141.68, 134.57, 134.46, 131.97, 131.63, 131.50, 130.71, 128.15, 127.71, 126.79, 126.56,119.80, 119.64, 119.33, 61.79, 58.05, 55.66, 54.41, 47.48, 46.35, 44.82, 38.00, 36.45, 31.93, 30.06, 29.65, 28.60. UV-vis [CH_2_Cl_2_, λ_max_ nm (ε, L·mol^−1^·cm^−1^), 5.90 × 10^−5^ M]: 418 (410,000), 516 (14,000), 549 (5900), 591 (3800), 647 (2600). HRMS (ESI^+^): *m/z* calcd. for MH^+^ [C_53_H_53_N_6_O_2_] = 805.4225, found 805.4215.

#### 3.3.2. General Procedure for the Sulfonation/Deprotection of the Mixed Porphyrins

In a round-bottomed flask, equipped with magnetic stirrer and a Dimroth reflux condenser capped with a calcium chloride tube, a stirred solution of the mixed porphyrin (1.0 mol equiv.) in concentrated (96% *w/w*) H_2_SO_4_ (20 mL/mmol porphyrin) was heated to 100 °C for 6H. After stirring for 18 h at rt, water (2 mL/mL H_2_SO_4_) was carefully added dropwise and the resulting dark-green suspension was centrifuged at 6000 rpm during 30 min. The supernatant was decanted, and the remaining sulfuric acid was neutralized with solid Na_2_CO_3_ to afford a purple-colored solution. Inorganic salts were removed by medium-pressure reverse-phase column chromatography using MCI GEL CHP20P 75–150 μm (Diaion^®^, Supelco/Sigma-Aldrich, Billerica, MA, USA), in which the porphyrin was slightly retained, thus allowing the elimination of inorganic salts using water as the eluent; increasing gradients of methanol (from 0% to 50%) conveniently eluted the porphyrin. The concentration of the sample by evaporation of the solvents under reduced pressure followed by lyophilization afforded the sodium salt of the sulfonated porphyrin as a purple-colored solid [46].

*(S)-5-(Pyrrolidin-1-ium-2-ylmethyl)-10,15,20-tris(4-sulfonatophenyl)porphyrin disodium salt***8**. Obtained in 82% yield (0.18 g, 0.19 mmol) from (*S*)-5-[(*N*-Boc-pyrrolidin-2-yl)methyl]-10,15,20-triphenylporphyirin **7** (0.17 g, 0.23 mmol) as a purple-colored solid. ^1^H-NMR (400 MHz, DMSO-*d_6_*, TMS_int_): δ (ppm) = 9.94–9.85 (m, 2H), 8.99–8.91 (m, 2H), 8.86–8.76 (m, 4H), 8.26–8.12 (m, 6H), 8.11–8.00 (m, 6H), 5.62–5.52 (m, 2H), 5.47–5.35 (m, 1H), 2.13–2.04 (m, 2H), 1.87–1.76 (m, 2H), 1.70–1.60 (m, 2H), −2.96 (br, 2H). Due to aggregation, only poorly resolved ^13^C-NMR spectra could be obtained for this compound. FTIR (ATR): ν = 3426, 1624, 1392, 1180, 1122, 1050, 1011,736, 632 cm^−1^. UV-vis [H_2_O, λ_max_ nm (ε, L·mol^−1^·cm^−1^), 1.25 10^−6^ M]: 413 (408,000), 517 (18,000), 554 (9200), 583 (8200), 634 (5200). HRMS (ESI^−^): *m/z* calcd. for M^3–^ [C_43_H_32_N_5_O_9_S_3_] = 286.0454 (*z* = 3), found 286.0455.

*(S)-5-[1-(Pyrrolidin-1-ium-2-ylmethyl)piperidin-1-ium-4-yl]-10,15,20-tris(4-sulfonatophenyl)porphyrin sodium salt***13**. Obtained in a 58% yield (70 mg, 0.07 mmol) from (*S*)-5-[1-*N-*Boc-2-(methylpyrrolidinyl)piperidin-4-yl]-10,15,20-triphenylporphyrin **12** (95 mg, 0.12 mmol) as a purple-colored solid. No satisfactory NMR spectra could be obtained for this compound. UV-vis [H_2_O, λ_max_ nm (ε, L·mol^−1^·cm^−1^), 5.90 × 10^−5^ M]: 413 (332,000), 521 (15,700), 559 (9500), 587 (7200), 646 (5100). HRMS (ESI^−^): *m/z* calcd. for M^3−^ [C_48_H_41_N_6_O_9_S_3_] = 313.7371 (*z* = 3), found 313.7372.

*(S)-(1-(Pyrrolidin-2-ylmethyl)piperidin-4-yl)methanol* (**17**). To a cold (0 °C) stirred solution of *N*-Boc-(*S*)-2-[(4-(hydroxymethyl)piperidin-1-yl)methyl]pyrrolidine **10** (100 mg, 0.33 mmol) in dry dichloromethane (5 mL), trifluoroacetic acid (5 mL) was added dropwise, and the resulting mixture was stirred at rt for 30 min. The volatiles were removed under vacuum, and the residue was diluted with methanol (10 mL). This process was repeated twice, and the residue was taken up in aqueous 1 M NaOH solution (20 mL). Extraction with ethyl acetate (3 × 10 mL), drying over sodium sulphate, and elimination of the solvents by rotary evaporation afforded the desired product 17 (48 mg, 0.24 mmol, 73% yield) as a yellowish oil, which was used for the catalysis experiments without further purification.

^1^H-NMR (400 MHz, D_2_O): δ (ppm) = 3.67−3.57 (m, 1H), 3.47 (d, *J* = 6.5 Hz, 2H), 3.20–3.10 (m, 2H), 3.01 (dd, *J* = 27.6 Hz, *J’* = 11.2 Hz, 2H), 2.64 (d, *J* = 6.1 Hz, 2H), 2.27–2.06 (m, 3H), 2.02–1.83 (m, 2H), 1.75 (d, *J* = 12.7 Hz, 2H), 1.64–1.51 (m, 2H), 1.34 (d, *J* = 6.5 Hz, 2H), 1.31–1.16 (m, 2H). HRMS (ESI^+^): *m/z* calcd. for MH^+^ [C_11_H_23_N_2_O] = 199.1805, found 199.1802.

#### 3.3.3. General Procedure for the Aqueous Organocatalysis of Aldol Reactions with the Amphiphilic Porphyrin-Amine Hybrids

A solution of the sulfonated amine-porphyrin hybrid (0.01 mmol) in water (Milli-Q, 1 mL) in a 10-mL round-bottomed flask was stirred at rt for 2 min; subsequently, the donor ketone (0.50 mmol) and 4-nitrobenzaldehyde (0.10 mmol) were added sequentially, and the resulting suspension was stirred (open air conditions) at rt until complete consumption of the aldehyde (TLC monitoring). After the addition of more water (10 mL), the reaction mixture was extracted with dichloromethane (3 × 10 mL). The combined organic phase was dried over anhydrous Na_2_SO_4_, and after filtration, the organic solvent was eliminated in vacuo. Finally, the residue was purified by column chromatography in silica gel, eluting with 1:1 hexane/ethyl acetate [47].

*2-(Hydroxy(4-nitrophenyl)methyl)cyclohexan-1-one* (**16a**, *anti* + **16b**, *syn*) [48]. Yellow-colored solid. ^1^H-NMR (400 MHz, CDCl_3_, TMS_int_): δ (ppm) = 8.25–8.18 (m, 2H*_anti_*, 2H*_syn_*), 7.54–7.47(m, 2H*_anti_*, 2H*_syn_*), 5.49 (t, *J* = 2.6 Hz, 1H*_syn_*), 4.90 (dd, *J* = 8.3 Hz, *J’* = 2.6 Hz, 1H*_anti_*), 4.06 (d, *J* = 3.0 Hz, 1H*_anti_*), 3.15 (d, *J* = 3.0 Hz, 1H*_syn_*), 2.67–2.31 (m, 2H*_anti_*, 2H*_syn_*), 2.16–2.07 (m, 1H*_anti_*, 1H*_syn_*), 1.90–1.32 (m, 6H*_anti_*, 6H*_syn_*).

HPLC (Chiralpak© IC, 1mL·min^−1^, Hexane:IPA 95:5, λ = 270 nm): t_R_ = *syn*, 17.9 min (1’*R*,2*R*), 19.6 min (1’*S*,2*S*); *anti*, 22.4 min (1’*R*,2*S*), 27.9 min (1’*S*,2*R*).

*4-Hydroxy-4-(4-nitrophenyl)butan-2-one* (**19**) [16]. Yellow-colored solid. ^1^H-NMR (400 MHz, CDCl_3_, TMS_int_): δ (ppm) = 8.22 (d, *J* = 8.8 Hz, 2H), 7.54 (d, *J* = 8.7 Hz, 2H), 5.27 (dd, *J* = 8.1Hz, *J’* = 4.1 Hz, 1H), 3.55 (br s, 1H), 2.87–2.83 (m, 2H), 2.22 (s, 3H).

HPLC (Chiralpak© IC, 1 × 250 mm × 4.6 mm, 1mL·min^−1^, Hexane:IPA 95:5, λ = 270 nm)**:** t_R_ = 33.0 min (*S*), 35.4 min (*R*).

#### 3.3.4. General Procedure for the Aqueous Organocatalysis of the Michael Addition of Cyclohexanone to 2-Nitrostyrene with the Amphiphilic Porphyrin-Amine Hybrids

A solution of the sulfonated amine-porphyrin hybrid (0.01 mmol) in a 1.5:1 water/THF mixture (0.20 mL) in a 10-mL round-bottomed flask was stirred at rt for 2 min; subsequently, cyclohexanone 14 (49 mg, 0.50 mmol) and 2-nitrostyrene 20 (15 mg, 0.10 mmol) were added sequentially, and the resulting suspension was stirred (open air conditions) at rt until complete consumption of the 2-nitrostyrene (TLC monitoring). After the addition of more water (7 mL), the reaction mixture was extracted with dichloromethane (3 × 10 mL). The combined organic phase was dried over anhydrous Na_2_SO_4_, and after filtration, the organic solvent was eliminated in vacuo. Finally, the residue was purified by column chromatography in silica gel, eluting with 6:1 hexane/ethyl acetate.

*(1’R*,2S*)-2-(2-Nitro-1-phenylethyl)cyclohexan-1-one* (**21**) [44]. Yellow-colored oil. ^1^H-NMR (400 MHz, CDCl_3_, TMS_int_): δ (ppm) = 7.36–7.23 (m, 3H), 7.19–7.13 (m, 2H), 4.94 (dd, *J* = 12.5 Hz, *J’* = 4.5 Hz, 1H), 4.64 (dd, *J* = 12.5 Hz, *J’* = 9.9 Hz, 1H), 3.76 (td, *J* = 9.9 Hz, *J’* = 4.5 Hz, 1H*_syn_*)*, 2.74 − 2.64 (m, 1H), 2.52 − 2.33 (m, 2H), 2.13-2.03 (m, 1H), 1.83–1.57 (m, 4H), 1.30–1.18 (m, 1H).

*The minor *anti*-isomer could be separated by chromatography and identified by its characteristic signal at 4.04–3.97 (m, 1H*_anti_*).

HPLC (Phenomenex i-cellulose-5, 1 × 250 mm × 4.6 mm, 1 mL·min^−1^, Hexane:IPA 90:10, λ = 220 nm): *_syn_*t_R_ = 37.8 min (1’*R*)-2-(*S*), 41.8 min (1’*S*)-2-(*R*).

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
