# Peer review of "Chiral Amphiphilic Secondary Amine-Porphyrin Hybrids for Aqueous Organocatalysis"

_molecules, 2020, doi:10.3390/molecules25153420_

Round 1

Reviewer 1 Report

The present paper describes the use of chiral organocatalysts based on amine-porphyrin systems in aldol and Michael reactions performed in water. Research on carrying out organic reactions in water systems is still a very relevant topic in modern organic chemistry, especially in terms of asymmetric synthesis. For this reason, I find the topic of research undertaken as very interesting.

The work is free from substantive errors, I only found some editorial mistakes (e.g. word repetition - page 6 line 125, page 6 line 128; typing error in the compound 11 name - Supplementary Material - page ESI 7). Thereby, I recommend publication this contribution in Molecules.

Author Response

The minor changes suggested by the reviewer (avoid word repetition in page 6, lines 125 and 128; corrected name of compound 11 "pyrrolidin-2-yl" in ESI-7) have been effected. 

Please note that after revision compound 11 is now compound 7

Reviewer 2 Report

In this paper, the authors reported the preparation and catalytic activity of the chiral secondary amine-porphirin hybrids, and that the molecules act as good catalyst in aqueous aldol addition reactions.

The authors emphasize that the catalytic activity of the molecules can be controlled by pH-dependent supramolecule behavior (aggregation/deaggregation state). They showed that the catalysts can be easily separated from reaction products by using the aggregation property of the molecules in acid condition.

The characterization of the molecules and catalytic tests were performed suitably.

I think that the results reported in this paper would be contributed to the development of the switchable catalysis.

There are minor comments as follows.

  1. Line 153,177

5 → 2

  1. Table 3 Catalyst

11 → 12

16 → 17

20 → 21

  1. Line 440,454

23 → 23

25 → 25

  1. Table 1

I would like to know the behavior of the catalyst 12 at low pH, although it may be obvious.

  1. Line 167

If the reaction mechanism of the catalyst 17 is described, the discussion would become rather clear.

Author Response

We thank the referee for his/her careful reading of the manuscript. The minor changes suggested have been made in lines 154, 155, 440, 454, and in Table 3 (catalyst numbers).

4. I would like to know the behavior of the catalyst 12 at low pH, although it may be obvious.

We have added a short paragraph just before Table 1 in which we specify that 12 affords a highly insoluble J-aggregate at pH below 4.3. A quantitative experiment however was performed only with catalyst 17

5. Mechanism of catalyst 17

As discussed in the last paragraph of page 7, we believe that an "in water" enamine mechanism (i.e., without hydrogen bonding between the aldehyde and the catalyst) is at work for our amphiphilic porphyrin hybrids.

Please note that after revision compound numbering has changed, 12 is now 8 and 17 is now 13

Reviewer 3 Report

This manuscript deals with the synthesis of two chiral, amphiphilic 5-substituted-10,15,20-tris(4-sulfonatophenyl)porphyrins containing (S)-proline-based moiety, and their pH-dependent supramolecular behaviour, as well as their use in some stereoselective model reaction as organocatalysts. Since the catalytic reactions and applying organocatalysis have high impact on the selective syntheses of organic compounds, this investigation deserves attention.

In this form, however, this manuscript cannot be published for the following reasons:

1) ps. 2–3 line 54 and 70  The Schemes 1 and 2 are practically the same as published in the authors’ previous work (Arlegui, A.; Torres, P., Cuesta, V.; Crusats, J., Moyano, A. pH-Switchable Aqueous Organocatalysis with Amphiphilic Secondary Amine–Porphyrin Hybrids. Eur. J. Org. Chem., doi: 10.1002/ejoc.202000648), and thus they do not give new information. It should be omitted them.

2) p. 7 line 171  It is difficult to understand the data of the Table 1 and follow your explanations, because it contains too many footnotes, and it is not clear which stereocenter is mentioned (2S). It would be more advantageous if the stereogenic centers were marked in the Scheme 7. Furthemore, the solvent was water in all cases, therefore it is unnecessary to write it in all rows. Similar ascertainments can be made regarding Tables 2 and 3. You should reorganise them.

3) ps. 11–13  During the characterisation of new compounds no melting points are given for the solid materials. Although they are characterised by 1H and 13C NMR, FTIR, HRMS and UV-Vis measurements, this traditional technique can provide useful information about these substances.

In addition, you should give more details about the Resolve C18, Nucleosil 120-5C18, Daicel Chiralpak® IC or Phenomenex® i-cellulose-5 HPLC columns, for example, in the following way: Resolve C18 column (X × Y mm, Z mm).

4) The English also needs some improvements. There are some typical grammar or typing mistakes:

p. 1 line 15 and elsewhere „… in acidic conditions …” instead of under acidic conditions

       line 19 and elsewhere„… can be easily separated ...”  instead of    can easily be separated

       line 41 „…   typically at values below 4.8, …”  instead of   typically below 4.8,

p. 2 line 58 „… can be readily prepared ...” instead of  can readily be prepared

       line 63 „After some experimentation …”  instead of   After some experiments

p. 5 line 109 „… 4-hydroxymethylpiperidine ...” instead of 4-(hydroxymethyl)piperidine

p. 6 line 157 „As it can be seen ...” instead of As seen

p. 7 line 162 „… had been …” instead of was

       line 171 entry 4 „… (2S) …”  instead of  (2S)

p. 8 line 203 „Aminoalcohol …” instead of Amino alcohol

       line 204 „… small catalytic activity …”  instead of  low catalytic activity

       line 213 „… has also been studied in aqueous media (Scheme 10), and best results have been …”  instead of  was also studied in aqueous media (Scheme 10), and the best results were

p. 11 line 304 „… 4-formyl-piperidin-1-yl ...” instead of 4-formylpiperidin-1-yl

p. 12 line 392 „… [(Pyrrolidin-1-ium-2-yl)methyl] ...” instead of (Pyrrolidin-1-ium-2-ylmethyl)

Author Response

We thank this referee for his/her attentive reading of the manuscript.

1) ps. 2–3 line 54 and 70  The Schemes 1 and 2 are practically the same as published in the authors’ previous work and they do not give new information.

We agree with the referee. We have therefore deleted Scheme 2 (with the subsequent renumbering of Schemes and compounds throughout the manuscript). We prefer to maintain Scheme 1 however, since we believe that it summarizes the basic concept of the research and can be useful to the reader without the need of a detailed perusal of our previous work (ref. 14). 

2) p. 7 line 171  It is difficult to understand the data of the Table 1 and follow your explanations, because it contains too many footnotes, and it is not clear which stereocenter is mentioned (2S). It would be more advantageous if the stereogenic centers were marked in the Scheme 7. Furthemore, the solvent was water in all cases, therefore it is unnecessary to write it in all rows. Similar ascertainments can be made regarding Tables 2 and 3. You should reorganise them.

We have depicted the absolute configurations of the major enantiomers of the aldol adduct in Scheme 6 (former Scheme 7). We have simplified the Tables 1-3  by eliminating the solvent column, as well as some footnotes.

3) ps. 11–13  During the characterisation of new compounds no melting points are given for the solid materials.

Porphyrins 7 and 12 were obtained as waxy solids, and sodium salts 8 and 13 decomposed before melting. Details on the HPLC columns have been added.

4) The English also needs some improvements. There are some typical grammar or typing mistakes.

The necessary corrections have been made.